# CAUSAL PROBABILISTIC SPATIO-TEMPORAL FUSION TRANSFORMERS IN TWO-SIDED RIDE-HAILING MARKETS

## ABSTRACT

Achieving accurate spatio-temporal predictions in large-scale systems is extremely valuable in many real-world applications, such as weather forecasts, retail forecasting, and urban traffic forecasting. So far, most existing methods for multi-horizon, multi-task and multi-target predictions select important predicting variables via their correlations with responses of interest, and thus it is highly possible that many forecasting models generated from those methods are not causal, leading to poor interpretability. The aim of this paper is to develop a collaborative causal spatio-temporal fusion transformer, named `CausalTrans`, to establish the collaborative causal effects of predictors on multiple forecasting targets, such as supply and demand in ride-sharing platforms. Specifically, we integrate the *causal attention* with the Conditional Average Treatment Effect (CATE) estimation method in causal inference. Moreover, we propose a novel and fast multi-head attention evolved from Taylor's expansion instead of *softmax*, reducing time complexity from $O(\mathcal{V}^2)$ to $O(\mathcal{V})$, where $\mathcal{V}$ is the number of nodes in a graph. We further design a spatial graph fusion mechanism to significantly reduce the parameters' scale. We conduct a wide range of experiments to demonstrate the interpretability of *causal attention*, the effectiveness of various model components, and the time efficiency of our `CausalTrans`. As shown in these experiments, our `CausalTrans` framework can achieve up to 15% error reduction compared with various baseline methods.

## 1 INTRODUCTION

This paper is motivated by solving a collaborative probabilistic forecasting problem of both supply and demand in two-sided ride-hailing platforms, such as Uber and DiDi. Collaborative supply and demand relationships are common in various two-sided markets, such as Amazon, Airbnb, and eBay. We consider two-sided ride-hailing platforms as an example. In this case, we denote supply and demand as online driver number and call orders, respectively, on the platform at a specific time in a city. Some major factors for demand include rush hours, weekdays, weather conditions, transportation network, points of interest, and holidays. For instance, if it rains during peak hours in weekdays, demand will dramatically increase and last for a certain time period. In contrast, some major factors for supply include weather, holidays, traffic condition, weekdays, and platform's dispatching and repositioning policies. Moreover, supply tends to gradually cover the area with many unsatisfied orders, that is, the distribution of supply tends to match with that of demand.

We are interested in establishing collaborative causal forecasting models for demand and supply by using various predictors (or covariates). Although many learning methods have been developed to address various collaborative prediction tasks, such as spatio-temporal traffic flow prediction (Zhu & Laptev, 2017; Du et al., 2018; Zhang et al., 2019b; Ermagun & Levinson, 2018; Luo et al., 2019), multivariate prediction (Bahadori et al., 2014; Liang et al., 2018), multi-task prediction (Tang et al., 2018; Chen et al., 2018; Chandra et al.,

2017), multi-view prediction (Yao et al., 2018), and multi-horizon prediction (Lim et al., 2019; Yu et al., 2020), these existing methods primarily select important predictors via their correlations with responses, leading to many forecasting models with poor interpretability. In contrast, we propose `CausalTrans`: a Collaborative Spatio-temporal Fusion Transformer, that generates causal probabilistic multi-horizon forecasts. To the best of our knowledge, this is the first work that captures collaborative causal effects of external covariates on multiple forecasting targets. Building such models is not only essential to enhancing forecasting performance, but also helps the platform to utilize various platform policies to match the distribution of supply with that of demand in two-sided markets.

In the `CausalTrans` framework, our major contributions are summarized as follows:

- We design the *causal attention* based on double machine learning (Chernozhukov et al., 2018) with two layers fully connected neural networks, and successful apply it to various large-scale time series forecasting problems. We conduct a wide range of experiments on real world datasets with multiple covariates and demonstrate that `CausalTrans` with *causal attention* outperforms many baseline models in various *Ride-hailing* scenarios.

- We propose a spatial fusion mechanism based on graph attention networks (`GAT`) (Veličković et al., 2017) to gather local regions and enhance robustness as adjacent regions always share similar supply and demand patterns.

- We propose an approximate time-efficient `Taylor` expansion attention to replace *softmax* in multi-head attention of Transformers (Vaswani et al., 2017) such that time complexity reduces from $O(\mathcal{V}^2)$ to $O(\mathcal{V})$. We carry out two groups of experiments with three multi-heads and five multi-heads to verify such efficiency improvement.

## 2 RELATED WORK

There is a large body of literature on vehicle flow forecasting (Zhu & Laptev, 2017; Bahadori et al., 2014; Tang et al., 2018; Lim et al., 2019; Yao et al., 2018). We selectively review several major methods as follows. In Zhu & Laptev (2017), the time series forecasting task as a two-step procedure includes offline pre-training and online forecasting. The offline pre-training step is an encoder-decoder framework for compressing sequential features and extracting principal components, whereas the second step gives explainable prediction changes under external variables. Bahadori et al. (2014) proposed a unified low-rank tensor learning framework for multivariate spatio-temporal analysis by combining various attributes of spatio-temporal data including spatial clustering and shared variables structure. For multi-step traffic flow prediction, Tang et al. (2018) proposed a spatio-temporal multi-task collaborative learning model to extract and learn shared information among multiple prediction tasks collaboratively. For example, such model combines spatial features collected from offline observation stations and inherent information between blended time granularities. Lim et al. (2019) proposed a temporal fusion transformer (`TFT`) to capture temporal correlations at each position, which was similar to self-attention mechanism and expected to capture long-term and short-term dependencies. Yao et al. (2018) proposed a deep multi-view spatio-temporal network (DMVST-Net), including a speed viewpoint (modeling the correlation between historical and future demand by LSTM (Gers & Schmidhuber, 2001)), a spatial viewpoint (modeling local spatial correlation by CNN), and a contextual viewpoint (modeling regional correlations in local temporal patterns). Overall, all above methods improve time series fitting by learning and predicting correlations across multiple spatio-temporal perspectives, targets, and tasks.

However, those methods lack convincing interpretability of "how and to what extent external variables affect supply and demand". Achieving good demand forecasting involves not only historical demand targets, but also various current external variables (e.g., weather conditions, traffic conditions, holidays, and driver reposition). Those historical demand observations were affected by historical external factors, so the demand forecasting only based on correlation between variables is hardly convincing. Furthermore, supply forecasting

is empirically affected by the distribution of demand besides current external variables. Establishing causal relationship between (supply, demand) and multiple external variables is critically important for accurate supply and demand forecasting.

## 3 METHODOLOGY

We introduce the `CausalTrans` framework to efficiently establish the collaborative causal effects of multiple predictors on spatio-temporal supply and demand below.

### 3.1 COLLABORATIVE SUPPLY AND DEMAND FORECASTING

We consider all related observations including supply, demand, and external variables collected in a city. Each day is divided into 24 hour segments and a city is divided into non-overlapping hexagonal regions (side length ranges from 600 to 1000 meters). The complete data consists of demand $x_v(t) \in R$, supply $y_v(t) \in R$, and dynamic covariates $z_v(t) \in R^z$, where $t$ is a specific hour segment and $v \in \mathcal{V}$ is a specific hexagon of the set of hexagonal regions, denoted as $\mathcal{V}$. Dynamic covariates includes weather, holidays, social events, POI (Point Of Interests), and government policies. Weather features consist of temperature ($^\circ$C), rainfall ($mm$), wind level and PM2.5 ($mg/m^3$). Holiday features are represented by one-hot boolean vectors, including seasons, weekdays, and national and popular holidays, such as Christmas Day. POI features are represented by the number of various positions, including traffic stations, business districts, communities, hospitals and schools. More detail cases about collaborative supply and demand are provided in Appendix A.

The problem of interest is to use all available observations in $\{(x_v(:t), y_v(:t), z_v(:,t)), v \in \mathcal{V}\}$ to predict $\{(x_v(t+1:t+\tau_{max}), y_v(t+1:t+\tau_{max})), v \in \mathcal{V}\}$, where $\tau_{max}$ is a pre-specified time length, $x_v(t_1:t_2)$ and $y_v(t_1:t_2)$ are the demand and supply vectors starting from time point $t_1$ to time point $t_2$, and $x_v(:t_2)$ and $y_v(:t_2)$ are the demand and supply vectors starting from the earliest time point to time point $t$. The demand $x_v$ may depend on historical supply $y_v$ that happens several weeks (or even longer) ago. But in the latest several weeks (training period), based on our understanding of ride-sharing business, demand $x_v$ may be primarily influenced by its own recent historical patterns. Based on the above description, we formulate the learning problem of collaborative demand and supply forecasting as follows:

$$P(x_v(t+1:t+\tau_{max})|x_v(:t), z_v(:t+\tau_{max})), \tag{1}$$

$$P(y_v(t+1:t+\tau_{max})|y_v(:t), x_v(:t+\tau_{max}), z_v(:t+\tau_{max})), \tag{2}$$

where $P(\cdot|\cdot)$ is a conditional distribution. In (1), it is assumed that $x_v(t+1:t+\tau_{max})$ is primarily affected by historical demands in $x_v(:t)$ and external covariates in $z_v(:t+\tau_{max})$. Furthermore, in (2), it is assumed that future supplies in $y_v(t+1:t+\tau_{max})$ are primarily affected by historical supplies in $y_v(:t)$, demand patterns in $x_v(:t+\tau_{max})$, and external covariates in $z_v(:t+\tau_{max})$. Comparing (1) with (2), we assume that the distribution of supply during $[t+1, t+\tau_{\max}]$ is driven by the historical and current distributions of demand besides the historical information in $y_v(:t)$ and external covariates in $z_v(:t+\tau_{max})$.

### 3.2 PROBABILISTIC FORECASTING

Most time series forecasting methods produce deterministic values, whereas forecasting results might have large variation and were hardly robust due to the variation of covariates and training process. To enhance forecasting reliability, we adapt the quantile loss function with the *Poisson* distribution as our final optimization function [1]. Empirically, following (Salinas et al., 2020; Wen et al., 2017; Li et al., 2019; Lim et al., 2019), we choose three quantile points $q \in Q = \{10\%, 50\%, 90\%\}$, in which the gap between forecasting values at

---

[1]Ride-hailing supply and demand variables approximately follow with the *Poisson* distribution.

90% and 10% percentiles can be regarded as the confidence interval. Take demand $x_t$ forecasting at time point $t$ as an example, the final quantile loss function is given by

$$\mathcal{L}_Q = \sum_{x_t \in \Omega} \sum_{q \in Q} \sum_{\tau=1}^{\tau_{max}} \frac{\mathcal{QL}_q(x_t, \hat{x}_{t-\tau}^q)}{\mathcal{M} \cdot \tau_{max}}, \quad (3)$$

where $\mathcal{QL}_q(x_t, \hat{x}_t^q) = \{q - \mathbf{I}(x_t \le \hat{x}_t^q)\}(x_t - \hat{x}_t^q)$, $\Omega$ is the training dataset, $\tau_{max}$ is the maximum prediction step, and $\mathbf{I}(\cdot)$ is an indicator function. For a fair comparison, given the test dataset $\widetilde{\Omega}$, we employ $q$-risk (Salinas et al., 2020; Lim et al., 2019; Li et al., 2019), denoted as $\mathcal{R}_q$, to evaluate the risk level of each quantile point as follows:

$$\mathcal{R}_q = \frac{2 \sum_{x_t \in \tilde{\Omega}} \sum_{\tau=1}^{\tau_{max}} \mathcal{QL}_q(x_t, \hat{x}_{t-\tau}^q)}{\sum_{x_t \in \widetilde{\Omega}} \sum_{\tau=1}^{\tau_{max}} |x_t|}. \quad (4)$$

There are at least two advantages of using the quantile loss function. First, the quantile loss function is more robust and stable than the mean square error or the hinge loss, especially when forecasting targets have large variation. Second, we can modify external covariates to change the confidence interval of *causal attention* and analyze real-world cases.

### 3.3 CAUSAL TRANSFORMER FRAMEWORK

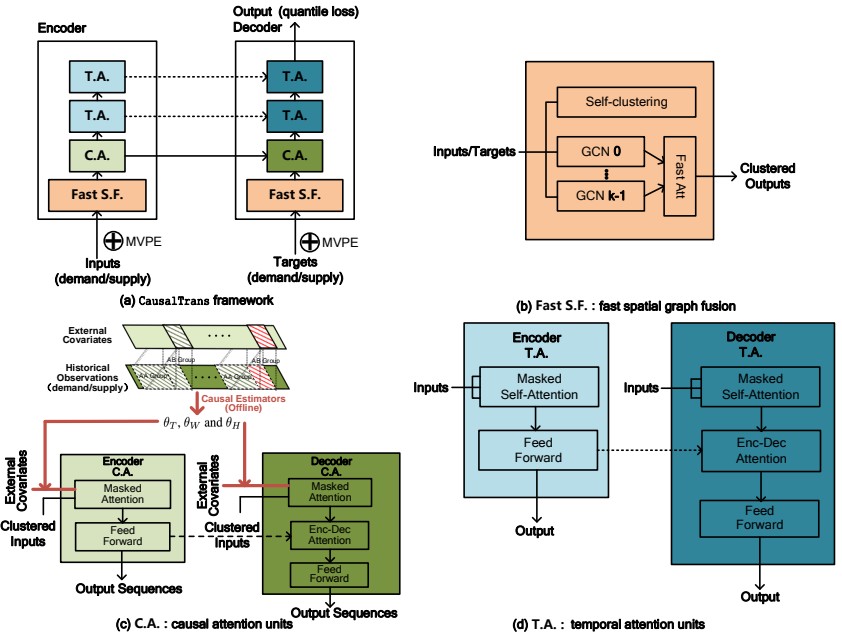

Figure 1: The overview of `CausalTrans` framework. Demand and supply are trained separately in sequence. (a). The framework consists of three essential components: *Fast S.F.* (*fast graph spatial fusion*), *C.A.* (*causal attention*), and *T.A.* (*temporal attention*). Moreover, we employ the average quantile loss distributed from {10%, 50%, 90%} to optimize forecasting probabilistic distributions. (b). The *Fast S.F.* consists of self-clustering with `GAT` and fast attention. (c). The *C.A.* applies offline trained causal weights $\theta$ to online treatments evaluations. (d). The *T.A.* aims to keep ordering self-attentions.

Our `CausalTrans` is a novel combination of causal estimators and the encoder-decoder architecture. Figure 1 shows the overview of the `CausalTrans` framework. The three key novel contributions of `CausalTrans` include *fast spatial graph fusion*, *causal attention*, and *temporal attention* units. First, from the spatial perspective, `CausalTrans` gathers a set of graph attention kernels (*GAT*) by using assignment scores extracted from temporal patterns. Moreover, we adapt the first-order Taylor's expansion on multi-head attention from transformer to reduce time complexity from square complexity to linear complexity. Second, from the temporal perspective, *causal attention* based on sufficient historical observations is trained offline to evaluate the causal weights on peek time slots, those on weather conditions, and those on holidays, which are denoted as $\theta_T$, $\theta_W$, and $\theta_H$, respectively, under diverse spatio-temporal conditions. Furthermore, we simplify three seasonal perspectives (week, month, and holidays) to represent multi-view position encoding (MVPE). Third, *temporal attention* is used to fill the gap between encoder and decoder, in which we add a sequence mask to ensure that the historical observations of time point $t$ only uses observations smaller than $t$. We set mask out to be $-\infty$ and illegal connection weights to be zero. In the following subsections, we introduce the main components of `CausalTrans`: fast spatial graph fusion and *causal attention* and show how they works together as a causal spatio-temporal predictor. Moreover, for notational simplicity, we focus on describing those components for forecasting demand $x_v$ in the following subsections, while avoid repeating the same components for supply $y_v$.

### 3.4 FAST SPATIAL GRAPH FUSION ATTENTION

In this subsection, we describe the fast graph fusion attention unit based on region clustering and fast multi-head attention. See Figure 1 (b) for the architecture of *Fast S.F.*. Since *GAT* has achieved impressive results in traffic forecasts (Park et al. (2019); Kosaraju et al. (2019); Zhang et al. (2019a)), we use *GAT* to extract contextual features in huge graphs. However, directly applying *GAT* to large-scale forecasting problems is a challenging task, so we design spatial fusion subgraphs that share local supply and demand information. Moreover, we build our framework based on transformers (Vaswani et al., 2017). Transformers have been state-of-the-art structure in various natural language processing (NLP) tasks (Wolf et al., 2019; Wang et al., 2019) and time series forecasting due to its prominent powers of long-term feature extraction and parallel computing. However, the multi-head attention in transformers becomes a key bottleneck for time efficiency. We design an approximate Taylor's expansion attention instead of using *softmax* function to accelerate matrix products. More detail results of fast attention can be found in Appendices C.2 and C.3.

We briefly describe the fast spatio-temporal fusion graph attention procedure below. First, let $X_t$ be the spatio-temporal demand feature matrix of all grids $\mathcal{V}$ before time $t$, the temporal patterns of $\mathcal{V}$ are represented as assignment scores given by

$$C = (c_{x_v,k}) = \overline{[\sigma_s(\sigma_r(X_t W_v) W_t)]}_{Batch}, \tag{5}$$

where $\overline{[\cdot]}_{Batch}$ is the mean operator on the batch mode, $k$ belongs to a $\mathcal{K}$-dimensional cluster vector, $v \in \mathcal{V}$, $W_v$ and $W_t$ are, respectively, spatial and temporal weight matrices corresponding to $X_t$, and $\sigma_s(\cdot)$ and $\sigma_r(\cdot)$ are *sigmoid* and *relu* activation functions, respectively.

Second, we use the $k$-th spatial learner $\mathcal{G}_k(x_v)$ to extract spatial features of sequential data $x_v$ in grid $v$, and the summed outputs of $\mathcal{K}$ clusters are given as follows:

$$h_v = \sum_{k \in \mathcal{K}} \mathcal{G}_k(x_v) c_{x_v,k}. \tag{6}$$

The softmax function is used to get attention weights among regions as follows:

$$\widehat{\alpha}_v = \sum_{v' \in \mathcal{N}_v} \alpha_{v,v'} \cdot x_{v'} = \frac{\sum_{v' \in \mathcal{N}_v} \exp(\sigma_r(\mathbf{a}^T[W \cdot x_v || W \cdot x_{v'}])) \cdot x_{v'}}{\sum_{v' \in \mathcal{N}_v} \exp(\sigma_r(\mathbf{a}^T[W \cdot x_v || W \cdot x_{v'}]))}, \tag{7}$$

where $\alpha_{v,v'}$ is the correlation weight between $v$ and $v'$, $\mathbf{a}$ and $W$ are network parameters, the superscript $T$ denotes the transpose of a vector or matrix, $\mathcal{N}_v = \{v'|v' \in \mathcal{V}, v' \neq v\}$ is the neighboring region set of region $v$, and $[\cdot||\cdot]$ is the concatenation operation.

In (7), the time complexity of computing $\exp(\sigma_r(\mathbf{a}^T[W \cdot x_v||W \cdot x_{v'}]))$ is $O(\mathcal{V}^2)$. Specifically, the exponent operation in $\exp(\mathbf{a}^T \cdot W) \cdot X$ of the softmax function limits the efficiency of attention. Moreover, cluster number $\mathcal{K} \ll \mathcal{V}$, and the time complexity of $\mathbf{a}^T W X$ is $O(\mathcal{K}^2 \cdot \mathcal{V}) \approx O(\mathcal{V})$. Many recent studies find that linear attention is feasible for tasks, whose primary focus is on short-term dependence. More details are discussed in Appendix D. Our novel linear attention is easy to implement and interpret. It follows from Taylor expansion that $\exp(\mathbf{a}^T W) \approx 1 + \mathbf{a}^T W$ under the condition of small $\mathbf{a}^T W$. Analogous to the self-attention in original Transformer (Vaswani et al., 2017), the approximate mean and variance of $\frac{QK^T}{\sqrt{d_k}}$ are 0 and 1, respectively, so $\mathbf{a}^T W$ here is limited to small values. We introduce $L_2$ normalization to ensure small $\mathbf{a}^T W$ and $1 + \mathbf{a}^T W \geq 0$ such that

$$\exp(\mathbf{a}^T W) \approx \mathcal{T}(\mathbf{a}^T W) = 1 + \left(\frac{\mathbf{a}}{||\mathbf{a}||_2}\right)^T \left(\frac{W}{||W||_2}\right). \tag{8}$$

where $\mathcal{T}$ is an approximate Taylor expansion. Equation (8) is close to inner dot products, which have advantages on parallel implementation and linear time complexity. Finally, $\widehat{\alpha}_v$ can be transformed into

$$\widehat{\alpha}_v = \frac{\sum_{v' \in \mathcal{N}_v} \mathcal{T}(\sigma_r(\mathbf{a}^T[W \cdot x_v||W \cdot x_{v'}])) \cdot x_{v'}}{\sum_{v' \in \mathcal{N}_v} \mathcal{T}(\sigma_r(\mathbf{a}^T[W \cdot x_v||W \cdot x_{v'}]))}. \tag{9}$$

### 3.5 CAUSAL ATTENTION MECHANISM

Many external covariates causally change the distribution of demand and supply as shown in Figures 2 and 3 of the supplementary document. Meanwhile, many existing works focus on finding the correlation between external covariates and forecasting targets. For example, Li et al. (2019) designed causal convolution to enhance the locality of attention, whereas Lim et al. (2019) added the variables selection networks and gate mechanism to train attention weights. These two studies (Lim et al., 2019; Li et al., 2019) intend to calculate correlations among variables, but not causal effects under counterfactual conditions. Statistically, such issue can be regarded as a heterogeneous treatment effect (*HTE*) problem. See Figure 1 (c) for the architecture of *C.A.*. To the best of our knowledge, *causal attention* methods for *HTE* have not been proposed in large-scale spatio-temporal forecasting problems.

First, we briefly describe the conditional average treatment effect (*CATE*) (Abrevaya et al., 2015). We still take demand vectors $x_v(t_1 : t_2)$ (abbreviated as $x$ in the following) of grid $v$ starting from time point $t_1$ to time point $t_2$ as an example. The $X$ represents a set of $x$. The treatments we consider include weather (rainfall, temperature and wind level), peek time slots and holidays. Let $x(s)$ be the target variable under treatment $s \in \mathcal{S}$, and $\mathbf{z}$ is a vector of other covariates. The *HTE* for comparing two treatment levels $s_0$ and $s_1$ is defined as

$$\tau(s_0, s_1; \mathbf{z}) = \mathbb{E}[X(s_1) - X(s_0)|\mathbf{z}]. \tag{10}$$

If treatment $s$ is continuous, then the treatment effect is defined to be $\mathbb{E}[\nabla_s X(s)|\mathbf{z}]$, where $\nabla_s = \partial/\partial s$.

To unbiasedly estimate treatment effects, we propose a *causal attention* module based on double machine learning (`DML`) (Chernozhukov et al., 2017) based on two layers non-parametric fully connected neural networks. Specifically, we assume

$$X(S) = \theta(\mathbf{z}) \cdot S + g_0(\mathbf{z}) + \epsilon \quad \text{and} \quad S = g_1(\mathbf{z}) + \eta, \tag{11}$$

where $\epsilon$ and $\eta$ are independent random variables such that $\mathbb{E}[\epsilon|\mathbf{z}] = \mathbb{E}[\eta|\mathbf{z}] = 0$, $g_0(\cdot)$ and $g_1(\cdot)$ are two non-parametric neural networks, and $\theta(\mathbf{z})$ is the constant marginal *CATE*. Let $\tilde{X} = X - \mathbb{E}(X|\mathbf{z})$ and

$\widetilde{S} = S - \mathbb{E}(S|\mathbf{z})$, we can get

$$\widetilde{X} = X - \mathbb{E}(X|\mathbf{z}) = \theta(\mathbf{z}) \cdot \{S - \mathbb{E}(S|\mathbf{z})\} + \epsilon = \theta(\mathbf{z}) \cdot \widetilde{S} + \epsilon. \qquad (12)$$

Therefore, we can compute $\theta(\mathbf{z})$ by solving

$$\hat{\theta}(\mathbf{z}) = \arg\min_{\theta \in \Theta} \mathbb{E}_n \left[ (\tilde{X} - \theta(\mathbf{z}) \cdot \tilde{S})^2 \right], \qquad (13)$$

where $\mathbb{E}_n$ denotes the empirical expectation.

Large historical data source contains all kinds of experimental environment and treatments. According to Algorithm 1, given time series $x_v(: t)$ at grid $v$ ($v$ is dropped for readability) and treatment $s_1 \in \mathcal{S}$, loop and search two treatment levels $s_0$ and $s_1$ along with the historical timeline to construct the *AB* groups $\{x(t_0)|s_0\}$ and $\{x(t_1)|s_1\}$. Then, we construct the *AA* groups $\{x(t_0 - \tau : t_0)\}$ and $\{x(t_1 - \tau : t_1)\}$ by a look-back window with the same length $\tau$ before $t_0$ and $t_1$, and make sure that both are both stationary processes with equal mean ($\mathcal{P}_{\texttt{KPSS}} > 0.05$ in $\texttt{KPSS}$ test (Shin & Schmidt, 1992) and $\mathcal{P}_{\texttt{T-Test}} > 0.05$ in $\texttt{T-Test}$ on both *AA* groups' first-order differences). Based on the selected *AA/AB* groups, we employ $\texttt{DML}$ to estimate *causal attention*. In our method, trained *causal attention* $\hat{\theta}$ will be inserted to transformer, and clustered regions share global $\hat{\theta}$ each other.

---

**Algorithm 1** Causal Attention Algorithm with $\texttt{DML}$

---

**Input:** Given demand matrix $x(: t)$ at a grid $v$ before time $t$, three kinds of treatments includes weekday and hour slots $T(: t) = \{W(: t), H(: t)\}$, weather vectors $W(: t)$, and holidays one-hot vectors $H(: t)$

**Output:** causal effect coefficients $\theta_T$ for $T(: t)$, $\theta_W$ for $W(: t)$, and $\theta_H$ for $H(: t)$

1: Take $\theta_T$ as an example, and suppose that a *AA* group and *AB* group on $T(: t)$ is $T_{AA} = T_{AB} = \{\}$
2: **for all** $\{T_w(t_0), T_w(t_1)\} \in \{Mon, Tue, ...Sun\}, \{T_h(t_0), T_h(t_1)\} \in \{1, ...24\}$ **do**
3:    **if** $T_w(t_0) = T_w(t_1), T_h(t_0) = T_h(t_1), \mathcal{P}_{\texttt{T-Test}}(x(t_0), x(t_1)) < 0.05$ **then**
4:       **for all** $t_0' \in \{: t_0\}$ and $t_1' \in \{: t_1\}$ **do**
5:          Calculate 1st-order differences $\widetilde{x}(t_0' : t_0)$ and $\widetilde{x}(t_1' : t_1)$
6:          **if** $\mathcal{P}_{\texttt{KPSS}}(\widetilde{x}(t_0' : t_0)), \mathcal{P}_{\texttt{KPSS}}(\widetilde{x}(t_1' : t_1))$ and $\mathcal{P}_{\texttt{T-Test}}(\widetilde{x}(t_0' : t_0), \widetilde{x}(t_1' : t_1)) > 0.05$ **then**
7:            $T_{AA}$.append($[(x(t_0' : t_0), x(t_1' : t_1))]$)
8:            $T_{AB}$.append($[(x(t_0), x(t_1))]$)
9:          **end if**
10:       **end for**
11:    **end if**
12: **end for**
13: Do $\texttt{DML}$ on $T_{AA}$ and $T_{AB}$ datasets and estimate treatment coefficients $\theta_T$
14: Repeat from Step 2 and estimate $\theta_W$ and $\theta_H$ by different $\texttt{DML}$.
15: **return** $\theta_T, \theta_W$, and $\theta_H$

---

## 4 EXPERIMENTS

### 4.1 DATASETS

We consider four datasets (*Electricity, Traffic, Retail*[2] *and Ride-hailing*) in our experiments as follows.

*Electricity. Electricity* contains hourly univariate electricity consumption of 370 customers. According to (Salinas et al., 2020), weekly oberservations before $t$ are inputs to predict the next 24 hours' series.

---

[2]https://www.kaggle.com/c/favorita-grocery-sales-forecasting/

Table 1: $\mathcal{R}_{50}/\mathcal{R}_{90}$ losses on the electricity and traffic datasets in the univariate group, where $\diamond$ denotes the results obtained from Li et al. (2019).

| | ARIMA | ETS | TRMF | DeepAR | DeepState | ConvTrans | Seq2Seq | MQRNN | TFT | CausalTrans |
|---|---|---|---|---|---|---|---|---|---|---|
| *Electricity* | 0.154/0.102$^\diamond$ | 0.101/0.077$^\diamond$ | 0.084/-$^\diamond$ | 0.075/0.040$^\diamond$ | 0.083/0.056$^\diamond$ | 0.059/0.034$^\diamond$ | 0.067/0.036$^\diamond$ | 0.077/0.036$^\diamond$ | **0.055/0.027**$^\diamond$ | 0.056/0.029 |
| *Traffic* | 0.223/0.137$^\diamond$ | 0.236/0.148$^\diamond$ | 0.186/-$^\diamond$ | 0.161/0.099$^\diamond$ | 0.167/0.113$^\diamond$ | 0.122/0.081$^\diamond$ | 0.105/0.075$^\diamond$ | 0.117/0.082$^\diamond$ | **0.095**/0.070$^\diamond$ | **0.095/0.065** |

Table 2: $\mathcal{R}_{50}$ losses on the retail and ride-hailing datasets. Percentages in brackets are loss reductions between CausalTrans and the second best result. $\diamond$ denotes results from Li et al. (2019).

| | ConvTrans | Seq2Seq | MQRNN | DeepAR | DMVST | ST-MGCN | TFT | CausalTrans |
|---|---|---|---|---|---|---|---|---|
| *Retail* | 0.429$^\diamond$ | 0.411$^\diamond$ | 0.379$^\diamond$ | 0.386 | 0.403 | 0.395 | 0.354$^\diamond$ | **0.352(-0.6%)** |
| *Ride-hailing* (1d, *city A*, Demand) | 0.573 | 0.550 | 0.495 | 0.499 | 0.524 | 0.482 | 0.450 | **0.434(-3.7%)** |
| *Ride-hailing* (1d, *city A*, Supply) | 0.482 | 0.453 | 0.428 | 0.422 | 0.443 | 0.421 | 0.415 | **0.393(-5.3%)** |
| *Ride-hailing* (1d, *city B*, Demand) | 0.470 | 0.455 | 0.405 | 0.400 | 0.422 | 0.404 | 0.370 | **0.361(-2.5%)** |
| *Ride-hailing* (1d, *city B*, Supply) | 0.426 | 0.404 | 0.388 | 0.384 | 0.388 | 0.378 | 0.357 | **0.341(-4.5%)** |
| *Ride-hailing* (7d, *city A*, Demand) | 0.756 | 0.717 | 0.653 | 0.663 | 0.664 | 0.677 | 0.689 | **0.613(-6.2%)** |
| *Ride-hailing* (7d, *city A*, Supply) | 0.612 | 0.569 | 0.516 | 0.519 | 0.536 | 0.575 | 0.583 | **0.468(-9.3%)** |
| *Ride-hailing* (7d, *city B*, Demand) | 0.693 | 0.627 | 0.574 | 0.571 | 0.590 | 0.588 | 0.576 | **0.539(-5.6%)** |
| *Ride-hailing* (7d, *city B*, Supply) | 0.568 | 0.519 | 0.499 | 0.501 | 0.503 | 0.525 | 0.528 | **0.454(-9.0%)** |

*Traffic. Traffic* contains hourly univariate occupancy rate of 963 San Francisco bay area freeways, where the look-back rolling window and prediction step are the same as *Electricity*.

*Retail. Retail* is the Favorita Grocery Sales Dataset from Kaggle competition (Lim et al., 2019), including daily metadata with diverse products, stores and external variables. To compared with some state-of-the-art methods (Lim et al., 2019; Salinas et al., 2020), historical observations across 90 days are trained to forecast product sales in the next 30 days.

*Ride-hailing.* The *Ride-hailing* dataset contains real supply, demand, and various of metadata at the hourly and hexagonal grid scale between June 2018 and June 2020 in two big cites (*city A* and *city B*) obtained from a ride-hailing company. The first 70%, the next 10% and the remaining 20% is used for training, validation and testing, respectively.

We group the first two datasets into the univariate group and the last two datasets into the multivariate group.

## 4.2 BENCHMARKS

In this section, two different forecasting methods, including iterative methods and multi-horizon methods, are compared in a wide range of comparison experiments. For our method CausalTrans, a pre-defined search space is used to determine optimal hyperparameters. Experimental details are included in Appendix B.

*Iterative methods.* Iterative methods generate multi-step prediction results by step-by-step rolling windows, where results in previous steps are used to as inputs in the next step. Typically, iterative methods include DeepAR[†], Deep State Space Models (DeepState[†]) (Rangapuram et al., 2018), ARIMA[†] (Zhang, 2003), ETS (Jain & Mallick, 2017) and TRMF (Yu et al., 2016).

*Multi-horizon methods.* Multi-horizon methods considered here include ConvTrans (Li et al., 2019), MQRNN[†] (Wen et al., 2017), Seq2Seq[†] (Sutskever et al., 2014), DMVST (Sutskever et al., 2014), ST-MGCN (Geng et al., 2019), and TFT (Lim et al., 2019). The † methods are trained by using the *GluonTS* (Alexandrov et al., 2019) package. DMVST and ST-MGCN are spatial baselines.

Table 3: $\mathcal{R}_{90}$ losses on the retail and ride-hailing datasets. Percentages in brackets are loss reductions between `CausalTrans` and the second best result. $\diamond$ denotes results from Li et al. (2019).

| | ConvTrans | Seq2Seq | MQRNN | DeepAR | DMVST | ST-MGCN | TFT | CausalTrans |
|---|---|---|---|---|---|---|---|---|
| *Retail* | $0.192^{\diamond}$ | $0.157^{\diamond}$ | $0.152^{\diamond}$ | 0.156 | 0.156 | 0.155 | $0.147^{\diamond}$ | **0.143(-2.8%)** |
| *Ride-hailing* (1d, *city A*, Demand) | 0.238 | 0.208 | 0.205 | 0.205 | 0.208 | 0.195 | 0.192 | **0.164(-14.6%)** |
| *Ride-hailing* (1d, *city A*, Supply) | 0.212 | 0.177 | 0.164 | 0.162 | 0.173 | 0.165 | 0.160 | **0.142(-11.3%)** |
| *Ride-hailing* (1d, *city B*, Demand) | 0.208 | 0.176 | 0.159 | 0.158 | 0.170 | 0.157 | 0.155 | **0.145(-6.5%)** |
| *Ride-hailing* (1d, *city B*, Supply) | 0.205 | 0.197 | 0.157 | 0.188 | 0.169 | 0.151 | 0.149 | **0.139(-6.7%)** |
| *Ride-hailing* (7d, *city A*, Demand) | 0.324 | 0.306 | 0.276 | 0.289 | 0.286 | 0.280 | 0.297 | **0.244(-11.6%)** |
| *Ride-hailing* (7d, *city A*, Supply) | 0.259 | 0.233 | 0.207 | 0.204 | 0.237 | 0.248 | 0.237 | **0.173(-15.2%)** |
| *Ride-hailing* (7d, *city B*, Demand) | 0.288 | 0.269 | 0.241 | 0.240 | 0.252 | 0.255 | 0.238 | **0.216(-9.3%)** |
| *Ride-hailing* (7d, *city B*, Supply) | 0.214 | 0.184 | 0.177 | 0.179 | 0.168 | 0.197 | 0.204 | **0.153(-8.9%)** |

### 4.3 RESULTS AND DISCUSSION

We adapt the quantile loss as optimization function, and compare various results by $q$-risk $\mathcal{R}_{50}/\mathcal{R}_{90}$ at quantile point $50\%/90\%$. More detailed descriptions of probabilistic forecasting are provided in subsection 3.2.

Table 1 includes the $\mathcal{R}_{50}/\mathcal{R}_{90}$ losses of all forecasting methods for *Electricity* and *Traffic* datasets. The *Electricity* data set does not have any covariates and is lack of spatial information, whereas the *Traffic* dataset does have spatial information even without multiple covariates. We observe that `ConvTrans` and `TFT` are comparable with each other and both outperform all other methods. We believe that compared with `TFT`, `ConvTrans` is able to take advantage of the spatial information in the *Traffic* dataset. This is not the case for the *Electricity* data set.

Table 2 and Table 3 include the $\mathcal{R}_{50}$ and $\mathcal{R}_{90}$ losses of all multi-horizon methods in the multivariate group. We consider both one-day and seven-day predictions and optimize the hyperparameters of all methods by using grid search. We have several important observations. First, for the one-day prediction, iterative `DeepAR` outperforms `Seq2Seq` and `MQRNN` due to the use of Poisson distribution and weather conditions. Second, for the spatial baselines `DMVST` and `ST-MGCN`, $\mathcal{R}_{50}$ and $\mathcal{R}_{90}$ losses are increasing with longer forecasting days, as such methods may overfit biased weights of external covariates. Third, `CausalTrans` outperforms all other competing methods primarily due to the use of the causal estimator `DML`. For instance, compared with the second best method, `CausalTrans` yields maximum 9.3% lower $\mathcal{R}_{50}$ and 15.2% lower $\mathcal{R}_{90}$ on the *Ride-hailing* (7d, *city A*, Supply) dataset. Fourth, `CausalTrans` achieves lower losses on forecasting supply than forecasting demand, since we explicitly model causal relationship between supply and demand in (2). Fifth, as expected, different with the one-day prediction, the seven-day prediction focuses on unbiased distribution estimation in order to alleviate error accumulation. This point of view is further reinforced by the results of the ablation study reported in Appendix C.2, and *causal attention* is visualized in Appendix C.1.

## 5 CONCLUSION

Based on causal inference theory, we develop the `CausalTrans` framework to address collaborative supply and demand forecasting in large-scale two-sided markets. We design the fast multi-head attention to improve the computational complexity to nearly linear $O(\mathcal{V})$. `CausalTrans` achieves similar performance as `TFT` based on the two datasets in the univariate group and outperforms all competing methods including `TFT` in the nine different experiments for the multivariate group. In particular, for our *Ride-hailing* datasets, `CausalTrans` can achieve up to 15% error reduction compared with various baseline methods. In the future, we will continue to integrate causal inferences with existing deep learning methods to deal with large-scale spatio-temporal forecasting problems.

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

## A    Ride-hailing Dataset Details

Taking *city A*[3] as an example, supply, demand, delta[4] and rainfall trends (January 1st, 2018 to January 1st, 2020) are plotted at daily scale in figure 2. We conclude that the variance of demand is bigger than supply, especially in raining rush hours.

Taking August 17th, 2018 in *city A* as another example in figure 3, we observe that the delta at dark red regions would not be for long, as spatio-temporal supply was changed by corresponding demand and reposition of drivers. The ride-hailing platform would release useful strategies to promote orders. Collaborative demand and supply implies that the distribution of supply corresponds to the distribution of demand.

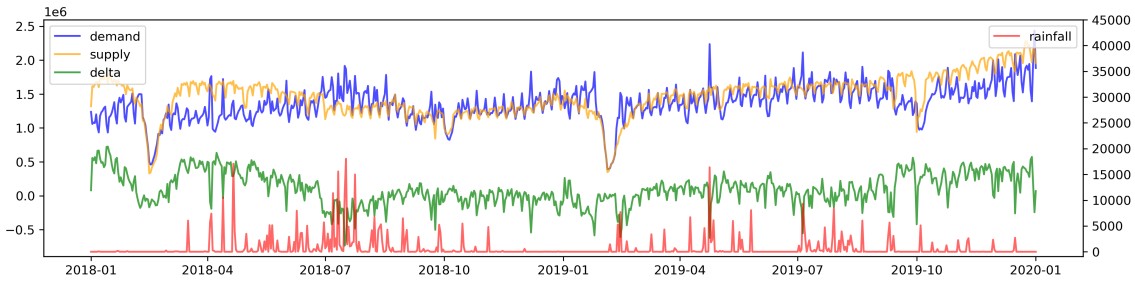

Figure 2: Supply, demand, delta and rainfall trends in *city A* ranging from January 1st, 2018 to January 1st, 2020. On August 17th, 2018 (Friday), demand increased significantly under heavy rains and evening peak hours. Lack of supply was far from being able to meet explosive demand. At the night peak hours on September 24th, 2018 (Mid-Autumn Festival), drivers' number was reduced and demand increased due to family reunion. During the National Days (from October 1st to 7th, 2018), commuting drivers and passengers both decreased, resulting in a bilateral decline of supply and demand. On December 7th, 2018 (Friday), heavy snow and low temperature stimulated potential demand. At the beginning of New Year's Eve in 2019, people were eager to reunite with families, resulting in low supply and high demand. After that, supply and demand tended to be balanced gradually.

## B    Training Details

Empirically, we consider determining optimal hyperparameters via a pre-defined random search space. For reproducibility, we include essential hyperparameters on our *Ride-hailing* dataset in Table 4.

Table 4: Optimal hyperparameters on *Ride-hailing* dataset.

|  | Learning rate | Dropout | Batch size | Num. multi-head | Sliding Window | Cluster $\mathcal{K}$ | Optimizer |
|---|---|---|---|---|---|---|---|
| *Ride-hailing* (1d, *city A*, Demand) | 0.001 | 0.3 | 64 | 3 days | 14 | 3 | Adam |
| *Ride-hailing* (1d, *city A*, Supply) | 0.001 | 0.3 | 64 | 3 days | 14 | 3 | Adam |
| *Ride-hailing* (1d, *city B*, Demand) | 0.001 | 0.3 | 128 | 4 days | 14 | 3 | Adam |
| *Ride-hailing* (1d, *city B*, Supply) | 0.001 | 0.3 | 128 | 4 days | 14 | 3 | Adam |
| *Ride-hailing* (7d, *city A*, Demand) | 0.001 | 0.1 | 512 | 5 days | 28 | 5 | Adam |
| *Ride-hailing* (7d, *city A*, Supply) | 0.001 | 0.1 | 512 | 5 days | 28 | 5 | Adam |
| *Ride-hailing* (7d, *city B*, Demand) | 0.001 | 0.1 | 512 | 5 days | 28 | 4 | Adam |
| *Ride-hailing* (7d, *city B*, Supply) | 0.001 | 0.1 | 512 | 5 days | 28 | 4 | Adam |

---

[3]*City A* is a big city in China.

[4]Supply minus demand is delta.

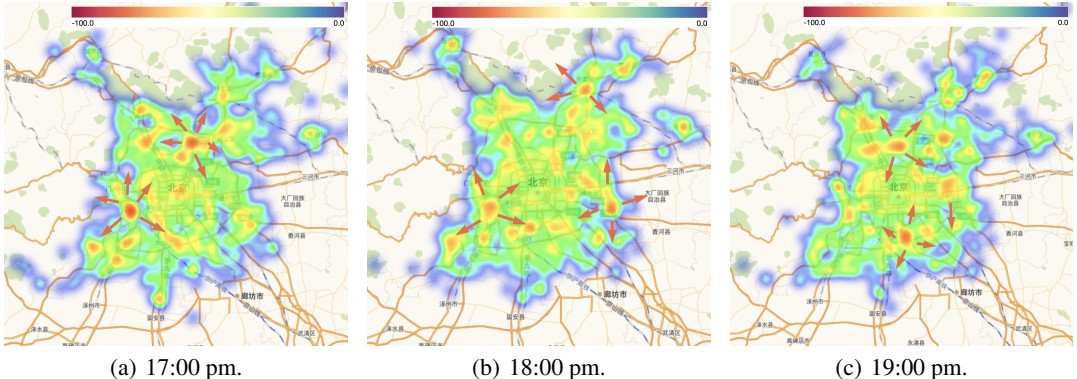

(a) 17:00 pm.      (b) 18:00 pm.      (c) 19:00 pm.

Figure 3: The delta heat maps at different o'clock of *city A* on August 17th, 2018. Panels (a), (b) and (c) show delta maps in peek hour 17:00pm, 18:00pm and 19:00pm, respectively. Delta values are normalized to (-100, 0) for data privacy, where -100 and 0 means the maximum and minimum delta value, respectively. As the evening peak process was going on, supply was matched with demand along the direction of red arrows in panels (a), (b) and (c) gradually.

## C    INTERPRETABILITY CASES

In this section, we analyze the impacts of essential components in `CausalTrans` and focus on what *causal attention* learns. First, since causal demand and supply are hardly assembled with unbiased estimation (Figure 2), we demonstrate attention-based interpretability in instance-specific significant events like frequent rainfall, holidays and peek time slots. Second, we perform ablation analysis about target probabilistic distribution `PoissonOutput`, *causal attention* with `DML` and `Uplift`, `FastAttention` and `SpatialFusion`. Finally, we compare fast improvements in multi-head attention on CPU (Intel Xeon E5-2630 2.20GHz) and GPU (Tesla P40), respectively.

### C.1    CAUSAL ATTENTION VISUALIZATION

As one of the most essential components, causal attention employs difference stationary tests and double machine learning to estimate coefficients $\theta(s)$ of treatment effects. In this section, we visualize *causal attention* distribution through sample-specific cases, including rainfall, weekdays, and time slots. Frequent rainfall is the most significant weather event for demand as described in Section A. Unlike with plenty of rainfall events, there are only a dozen of holidays in one year. If sequential context before one holiday fails to pass `Kpss` stationary test, causal estimator would not to be applied in training attention weights. Large-scale dataset is the fundamental to our method. For the diverse peek time slots, Section 3.1 concludes that demand and supply distributes different at commuting peeks and night hours. In addition, seasonal fluctuation and government's policies (e.g. traffic restriction in National Day) are considerable factors.

*Rainfall*. Take demand forecasting at an anonymous region in *city A* as an example, treatment is rainfall $s$, target is demand $x$, and other covariates $\mathbf{z}$ include regional id, time slots and holidays. For convenience, we select a group of adjacent *AB* groups from sufficient rainfall cases to give an interpretation. In Figure 4, we backtrack rainfall treatments to fix *AB Group 2*, and search *AB Group 1* by controlling similar covariates. Similarity means that both first-order differences are stationary, and then we construct a group of simple randomized controlled experiments. Given estimated $\theta(\mathbf{z})$ by running `DML`, we plot the distribution of *causal*

*attention* on the right side of green line. In practice, large amounts of increasing data would enhance the robustness of causal evaluation iteratively.

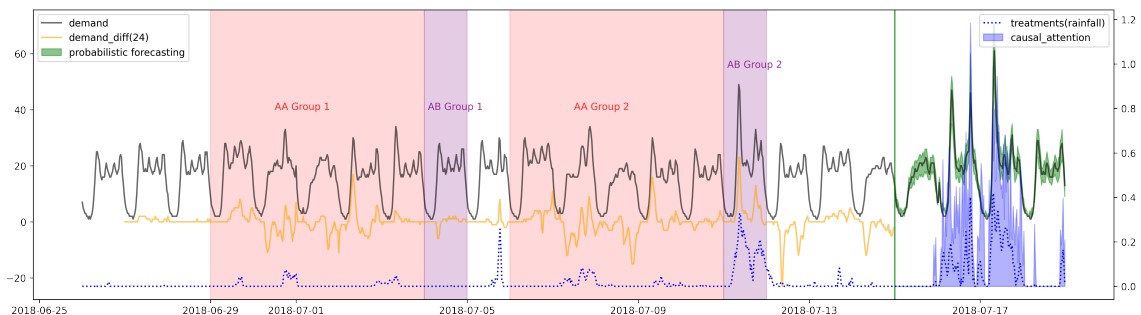

Figure 4: Causal attention and probabilistic demand forecasting treated by rainfall in some days at a anonymous region in *city A*. A time slice means an hour. The black solid line is real demand time series, and orange solid line is temporal differences in next 24 hour slices. The green vertical line means a starting point of forecasting, and subsequent green filled areas describe a confidence interval between quantile 10% and 90%. According to *causal attention*, "AA group 1" and "AA group 2" (two red filled areas) are regarded as comparable contexts, as the first-order difference of both groups passes `kpss`(Kwiatkowski et al., 1992) stationary test ($\mathcal{P} > 0.05$). "AB group 1" and "AB group 2" (two purple filled areas) is control group and treatment group, respectively. Based on homologous hypothetical controlled experiments, *causal attention* with future weather information would be added in forecasting to learn inferences.

*Collaborative demand and supply*. As described in Section 3.1 and equation (2), the distribution of supply is driven by the spatio-temporal patterns of demand. Similar with above *Rainfall* analysis, we take another anonymous region in *city A* as an example. In this case, forecasting target is supply, causal treatment is demand, and external variables include weather, time slots and holidays. According to Algorithm 1, our method needs to construct *AB* groups and corresponding lookback *AA* groups from large-scale historical data. For both *AB* controlled experiments, the average demands of AB and AA groups should be significant different, while supply is unlimited. In *AA* experiments, we empirically suggest that the time span maintains for at least one day. We trace back data to the past, but selected AA groups should satisfy randomization grouping hypothesis passed by *t-test*. Such periods with stable supply are abundant in recent years, which implies that we can easily find proper evaluation dataset for diverse regions. In Figure 5, trained *causal attention* demonstrates the demand's causal weights reflect in supply forecasting. Additionally, more novel causal modules similar with equation (2) can be designed to enhance interpretability and robustness, and such modules support end-to-end training in `CausalTrans` as well.

## C.2    ABLATION ANALYSIS

This subsection focuses on the performance of `CausalTrans` when some components are excluded. Proposed essential items contain tricky `PoissonOutput`, *Causal Attention*(`C.A.`), `FastAttention` and `SpatialFusion`. `C.A.` can be implemented by different causal algorithms, such as `DML` and `Uplift` (Künzel et al., 2019). As shown in Table 5, we list $\mathcal{R}_{50}$ (50% quantile point) losses on previous eight *Ride-hailing* datasets. Table 5 demonstrates that `C.A.`(`DML`) outperforms all of other components, and causal supply can be clearly influenced by causal demand. Finally, both `FastAttention` and `SpatialFusion` are not harmful to forecasting performance.

Furthermore, spatial fusion shows tiny improvement (+0.3% on average) in Table 5. We feel that spatial fusion aggregates adjacent hexagonal grids, leading to reducing statistical noises in both demand and supply. For

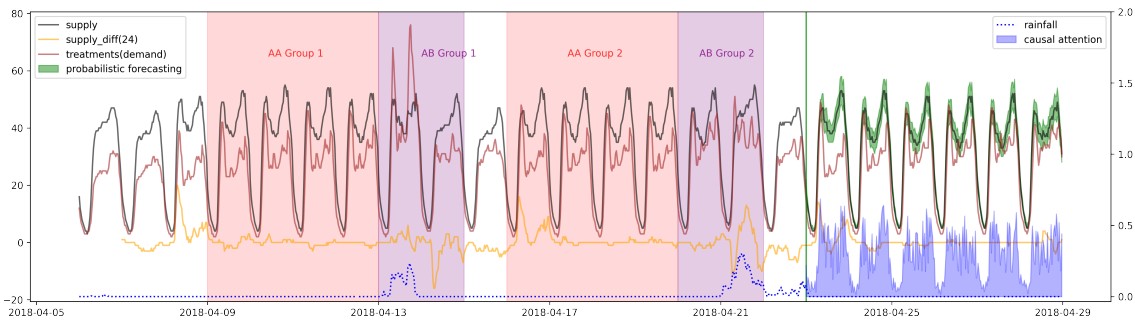

Figure 5: Causal attention and probabilistic supply forecasting treated by demand in some days at an anonymous region in *city A*. The black solid line is real supply time series, while other lines and areas are similar with Figure 4. *AA* group 1 and *AA* group 2 (two red filled areas) are selected by *causal attention*, and both *AA* groups have analogous *rainfall*, the same *weekday* and *time slots*. Mean difference of both *AA* groups in *t-test* is not significant. As *rainfall* can change demand and following demand changes supply, we estimate collaborative demand causal effects by adding *rainfall* causal effects. Fortunately, there are no *rainfall* and *holidays* factors in evaluation periods (right-hand side of the green vertical line), and therefore we can visualize the pure *causal attention* distributed in $(0, 1)$.

Table 5: Ablation analysis ($\mathcal{R}_{50}$ losses) of various components in `CausalTrans`. Each value represents the $\mathcal{R}_{50}$ loss that eliminates a specific component, and percentages in brackets are loss variations. For *Traffic* data set, `PoissonOutput` and `SpatialFusion` are positive components. For *Ride-hailing* data set, each component reflects different importance varying on different tasks. For `PoissonOutput`, forecasting demand and supply are not significantly different, but longer predict step leads to a bigger loss increment. Lack of `PoissonOutput` increases 2% loss on average. The most essential component `C.A.(DML)` improves collaborative supply forecasting more than demand. Long-term prediction depends on `C.A.(DML)` as well as the causal covariates. `C.A.(Uplift)` is similar to `C.A.(DML)`, but the simpler `C.A.(Uplift)` means less importance. `FastAttention` (-0.2% on average) and `SpatialFusion` (+0.3% on average) are proposed for reducing time complexity significantly without harming loss.

| | PoissonOutput | C.A.(DML) | C.A.(Uplift) | FastAttention | SpatialFusion |
|---|---|---|---|---|---|
| *Traffic* | 0.098(+3.1%) | / | / | 0.095(-0.4%) | 0.097(+2.5%) |
| *Ride-hailing* (1d, *city A*, Demand) | 0.444(+2.3%) | 0.463(+6.6%) | 0.457(+5.2%) | 0.433(-0.2%) | 0.431(-0.8%) |
| *Ride-hailing* (1d, *city A*, Supply) | 0.400(+1.7%) | 0.424(+7.9%) | 0.417(+6.1%) | 0.392(-0.3%) | 0.394(+0.2%) |
| *Ride-hailing* (1d, city B, Demand) | 0.366(+1.5%) | 0.381(+5.6%) | 0.379(+5.1%) | 0.361(+0.0%) | 0.363(+0.5%) |
| *Ride-hailing* (1d, city B, Supply) | 0.345(+1.1%) | 0.363(+6.4%) | 0.360(+5.5%) | 0.341(+0.1%) | 0.344(+0.8%) |
| *Ride-hailing* (7d, *city A*, Demand) | 0.633(+3.2%) | 0.674(+10.0%) | 0.660(+7.7%) | 0.611(-0.4%) | 0.612(-0.2%) |
| *Ride-hailing* (7d, *city A*, Supply) | 0.483(+3.3%) | 0.537(+14.8%) | 0.520(+11.1%) | 0.465(-0.6%) | 0.472(+0.9%) |
| *Ride-hailing* (7d, city B, Demand) | 0.549(+1.8%) | 0.584(+8.4%) | 0.575(+6.7%) | 0.540(+0.2%) | 0.546(+1.3%) |
| *Ride-hailing* (7d, city B, Supply) | 0.460(+1.3%) | 0.503(+10.7%) | 0.494(+8.8%) | 0.453(-0.3%) | 0.457(+0.7%) |
| **Average** | **+2.0%** | **+8.8%** | **+7.0%** | **-0.2%** | **+0.3%** |

instance, in some cases, the boundary (usually around 800 meters) of adjacent grids separates large demand hotpots (e.g., large shopping malls), resulting in some noise when counting supply and demand. Spatial fusion can reduce the influence of such noise, while improving the probabilistic forecasting performance. According to Table 5, the longer forecasting time (e.g., 7 days versus 1 day), the more significant gain by using spatial fusion. We consider the use of spatial fusion as a trick for enhancing the robustness of forecasting. The hyperparameter of spatial fusion is $\mathcal{K}$ used in the kmeans method. In this paper, we set $\mathcal{K} \in \{3, 4, 5\}$. More ablation analysis about $\mathcal{K}$ is shown in Table 6.

Table 6: Ablation analysis ($\mathcal{R}_{50}$ losses) of various cluster number $\mathcal{K}$ in `SpatialFusion`. Each value represents the $\mathcal{R}_{50}$ loss on a specific $\mathcal{K}$, and percentages in brackets are loss variations. Roughly speaking, seven-day prediction needs bigger $\mathcal{K}$ than one-day. We conclude that the longer prediction range needs more heterogeneous patterns. Optimal $\mathcal{K}$ on various *Ride-hailing* subdatasets are shown in Table 4.

|  | $\mathcal{K} = 2$ | $\mathcal{K} = 3$ | $\mathcal{K} = 4$ | $\mathcal{K} = 5$ | $\mathcal{K} = 6$ |
|---|---|---|---|---|---|
| *Ride-hailing* (1d, *city A*, Demand) | 0.442(+1.9%) | **0.434(+0.0%)** | 0.446(+2.8%) | 0.448(+3.3%) | 0.449(+3.4%) |
| *Ride-hailing* (7d, *city A*, Demand) | 0.641(+4.5%) | 0.630(+2.7%) | 0.617(+0.7%) | **0.613(+0.0%)** | 0.629(+2.6%) |

## C.3    TIME EFFICIENCY IMPROVEMENT

One of innovations proposed in this paper is to shorten running time of attention without losing overall quantile loss. The long experiment cycle suggests that we should choose a representative dataset, such as one-day demand prediction in *city A*. Data size of *city A* is large enough to reflect robust attention weights. In such dataset, we are only interested in the decrease of running time as the number of heads in multi-head attention decreases. As shown in Figure 6, when multi-head is 3, the reduction ratios of `CPU(20)`, `GPU(1)` and `GPU(2)` compared with softmax are 58%, 70%, and 68%, respectively. Similarly, when multi-head is equal to 5, the responding reduction ratios are, respectively, 49%, 58% and 60%. An exact time complexity is $O(\mathcal{K}^2 V)$ (see in Section 3.4), the smaller $\mathcal{K}$, the longer running time. In summary, proposed time-efficient attention outperforms default softmax attention significantly.

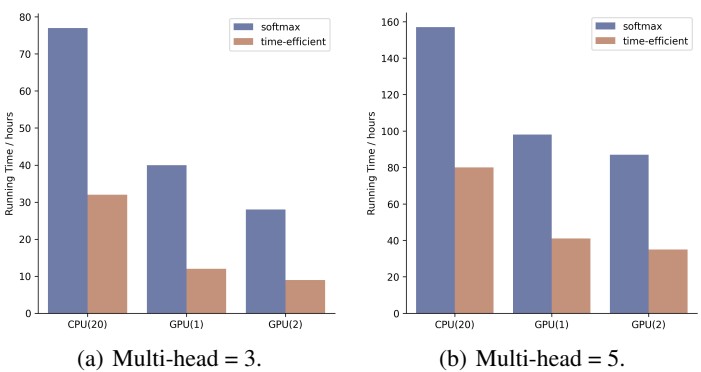

(a) Multi-head = 3.     (b) Multi-head = 5.

Figure 6: Time efficiency improvements on one-day demand prediction in *city A*. The numbers in brackets mean logic cores in a chip. `CPU(20)` utilizes multiple processing to accelerate matrix multiplication. The comparison of `GPU(1)` and `GPU(2)` aims to demonstrate a possibility of applying powerful GPUs in real world. Setting up (a) 3 heads and (b) 5 heads in multi-head attention is quite different. According to equation (9) and above bar plots, the less number of heads, the shorter running hours. Each running hour result is averaged by using three independent experiments.

## D    DISCUSSIONS ON LINEAR ATTENTION

In subsection 3.4, we propose a novel linear attention based on approximate Taylor expansion of exponential function. In contrast, other important methods are also developed to reduce attention cost. These attention acceleration methods can be roughly clarified into two groups. The first one is to construct kernel functions to

approximate softmax function, denoted as

$$\text{softmax}(Q^T K) = \varphi(Q)^T \cdot \phi(K), \tag{14}$$

where $Q$ and $K$ are query matrices and key matrices, respectively. For instance, Katharopoulos et al. (2020) construct a kernel function with basis function $\varphi(x) = \phi(x) = elu(x) + 1$ and reduce the computation complexity from $O(N^2)$ to $O(N)$, but such performance is just only concluded from image dataset. Shen et al. (2018) further explore a series of kernel forms to dissect Transformer's attention. They proposed a new variant of Transformer's attention by modeling the input as a product of symmetric kernels. This approach replaces the calculation order of softmax, which is equivalent to the basis function $\phi(x) = \text{softmax}(x)$ and $\varphi(x) = e^x$.

The second one is to modify attention's definition. Child et al. (2019) develope sparse factorizations of the attention matrix, which reduce the computation to $O(N\sqrt{N})$, but its attention hyperparameters are very hard to be initialized and actual efficiency is hard to ensure. Kitaev et al. (2020) propose Reformer to replace dot-product attention by one that uses locality-sensitive hashing, changing its complexity from $O(N^2)$ to $O(Nlog(N))$, where $N$ is the length of the sequence. Furthermore, they use reversible residual layers instead of standard residuals, allowing storing activations only once in the training process instead of $L$ times, where $L$ is the number of layers. However, Reformer is difficult to be implemented and applied in different tasks. Wang et al. (2020) demonstrate that the self-attention mechanism can be approximated by a low-rank matrix, and further propose Linformer mechanism to reduce the overall self-attention complexity to $O(N)$. Linformer uses two additional matrices $E$ and $V$ to project $K$ and $V$, respectively, in order to get $Attention(Q, K, V) = softmax(Q(EK)^T)FV$. But the MLM experiment in Linformer does not need to extract long-term dependence and cannot verify its linear time complexity for capturing long-term attention. Eliminating redundancy vectors from the self-attention is a key design idea. Furthermore, Goyal et al. (2020) exploit redundancy pertaining to word-vectors, and propose PoWER-BERT to achieve up to 4.5x reduction in inference time over BERT with <1% loss in accuracy on the standard GLUE benchmark. Similarly, Dai et al. (2020) propose Funnel-Transformer, which gradually compresses the sequence of hidden states to a shorter one, and hence reduces the computation cost. Finally, for our approximate Taylor expansion of softmax attention, if feature maps (i.e. $Q$, $K$ and $V$ in self-attention) meet the positive definite and normalization conditions and our task focuses on short-term dependence, then our linear attention would be useful for this aspect.

