# OpenReview forum: "Causal Probabilistic Spatio-temporal Fusion Transformers in Two-sided Ride-Hailing Markets"
_ICLR.cc/2021/Conference — Reject_

### Official Review · AnonReviewer3 · 2020-10-27
**Recommendation to Reject**

**Rating:** 2
**Confidence:** 5

**Review:**

The paper is interested with multivariate probabilistic forecasting applied in the context of ride-hailing forecast.
The goal is to be able to handle the spatial aspect, causal effects of external covariates (e.g. the causal impact of rain, Christmas on supply/demand) and dependency between supply and demand. To this end, the authors propose a causal attention mechanism	(to not just detect correlation with covariates) and a custom transformer architecture where the quadratic attention cost is avoided. Experiments are performed on public and private datasets against a set of (mostly) univariate baselines.

Strong points:
+ highly-relevant problem
+ novel consideration of handling causality instead of simple covariate correlation in this context

Weak points:
- very hard to read, many details missing and notation are not introduced
- lack of relevant baselines (e.g. baselines using spatial information)
- lack of relevant metrics to illustrate the benefit of the contribution
- missing related work discussion for efficient attention computation

I recommend a reject for this paper.

While the problem presented by the paper is highly relevant for the community, the paper has several issues that makes it not ready for publication. The first issue comes in the clarity of the description and in particular section 3.3 where many terms and notation are not introduced at all making the paper very hard to read (see detailed comments).

Experiments are also problematic as many details were unclear (see detailed comments) and would be far from being reproducible. But the most problematic bit is in its design: key aspects introduced in the paper are not asserted in the experiments.1) While the method claims ~5% error improvement on the ride-hailing benchmark, only one baseline have access to spatial information while numerous methods have been proposed to handle spatio-temporal forecast, only one baseline (TFT) is provided with spatial information (with no detail). 2) Since you are interested in the probabilistic forecast of the joint demand/supply targets (collaborative), your experimental setup should consider a joint metric (CRPS-sum for instance [1,2]) rather than the average of R50/R90 metrics of demand/supply targets which is "blind" to correlation between the two demand and supply targets.

Finally, I had issue with the Taylor expansion as its description was not clear (see detailed comments) but also I found description of related work missing in this aspect. Attention cost has been decreased to $N\sqrt{N}$, $N\log{N}$ and $N$ respectively in [3, 4, 5], this discussion is clearly missing to relate your contribution.

# Additional questions to the author
- 3.1 at no point in the paper you say explicitly what is the loss that you are minimizing, this makes it harder to read. I would recommend to specify in 3.1 directly that you minimize NLE with poisson distribution (which is what I understood) and specify also clearly how predictions are made.
- 3.3 the beginning of the section mix model description and related work
- 3.3 Eq. (3) has two unintroduced notation, what is the \bar? what is "batch"?
- 3.3 Eq. (6) is so confusing, you are mixing norms and vectors (unintroduced), $T$ is not explicitly introduced, there is no clear explanation why the approximation holds also (e.g. why a^T W would be small)
- 4.2 DeepState is *not* an auto-regressive model, it belongs to the second category.
- 4.3: "CausalTrans outperforms all other competing methods primarily due to the use of the causal estimator DML and spatial information". This is very problematic as you dont compare with methods designed to handle spatial information.

# Additional feedback (not part of the decision assessment)
- 3.1 eq (1) (2) makes an assumption that only demand impacts supply: it would be good to discuss it at least. One could imagine where it breaks.
- 3.1 weather features: you should precise whether they are known in advance
- 3.3 The section will be easier to read if you indicate variable dimensions

Finally, I would recommend to use a public dataset rather than the private one for your benchmark (e.g. NY taxi dataset or Uber), you also have covariates such as date or weather forecast and this would make your work comparable and reproducible in the future.



[1] High-dimensional multivariate forecasting with low-rank Gaussian Copula Processes http://papers.nips.cc/paper/8907-high-dimensional-multivariate-forecasting-with-low-rank-gaussian-copula-processes

[2] Multi-variate Probabilistic Time Series Forecasting via Conditioned Normalizing Flows https://arxiv.org/abs/2002.06103

[3] Generating Long Sequences with Sparse Transformers https://arxiv.org/pdf/1904.10509.pdf

[4] Transformers are RNNs: Fast Autoregressive Transformers with Linear Attention https://arxiv.org/abs/2006.16236

[5] Reformer: The Efficient Transformer https://arxiv.org/abs/2001.04451

---

> ### Author Response · Authors · 2020-11-20
> **Thank you very much for many insightful comments, and we plan to add more spatial baselines and discussion about linear attention.**
>
> 11. "3.3 Eq. (6) is so confusing, you are mixing norms and vectors (unintroduced), is not explicitly introduced, there is no clear explanation why the approximation holds also (e.g. why a^T W would be small)"
> Response: Thanks a lot for pointing out our mistake. Equation (6) should be corrected as: $\exp(a^TW) \approx \mathcal{T}(a^TW) = 1 + \left(\frac{a}{||a||_2}\right)^T \left(\frac{W}{||W||_2}\right)$。Analogous to the self-attention in original Transformer, the approximate mean and variance of $\frac{QK^T}{\sqrt{d_k}}$ are 0 and 1, respectively, so $a^TW$ here are limited to small values. We introduce $L_2$ normalization to ensure small $a^TW$ and $1 + a^TW \geq 0$, as mentioned in equation (6) above.
>
> 12. "4.2 DeepState is not an auto-regressive model, it belongs to the second category."
> Response: We tend to classify DeepState as an iterative step-by-step forecasting method (i.e., the first category), which is consistent with categories in TFT paper [1].
> [1]. Lim, Bryan, et al. "Temporal fusion transformers for interpretable multi-horizon time series forecasting." arXiv preprint arXiv:1912.09363 (2019).
>
> 13. "4.3: "CausalTrans outperforms all other competing methods primarily due to the use of the causal estimator DML and spatial information". This is very problematic as you don't compare with methods designed to handle spatial information."
> Response: As mentioned in the second comment above, we plan to add more spatial baselines as competitors.
>
> 14. "3.1 eq (1) (2) makes an assumption that only demand impacts supply: it would be good to discuss it at least. One could imagine where it breaks."
> Response: Thanks a lot! Equation (1) and (2) are primarily based on our understanding of business. The first one is that the demand $x$ may depend on historical supply $y$ that happens several weeks (or even longer) ago. But in the last several weeks (training period), for customers, their demands may be primarily influenced by their own historical patterns and their recent request completion rates. The second one is that the demand may rise when supply is not enough for demand and demand may accumulate.
> In short, we feel that the distribution of demand is primarily determined by its historical distribution since the forecast period that we consider in the ride-sharing business is a short-term range (usually 1~7 days). Regarding to the first comment, we usually include historical request completion rate as an external covariate in $z$ to explain it better. The second one usually happens during the rush hour and lasts for 1-2 hours, leading to 2-4 outlying observations due to the temporal resolution being 30 minutes. Moreover, we have included the historical and current demands to predict the future demands such that the proposed equations work reasonably well. To address this issue better, we plan to introduce the request completion rate and integrate it with existing external covariates.
>
> 15. "3.1 weather features: you should precise whether they are known in advance"
> Response: The future weather features (i.e. the weather forecast) are known conditions.
>
> 16. "3.3 The section will be easier to read if you indicate variable dimensions"
> Response: Thanks a lot! We consider to add more details about variable dimensions in Appendix.
>
> 17. "...I would recommend to use a public dataset rather than the private one for your benchmark..."
> Response: Thanks for your kindly suggestion. At the beginning of this work, we consider that the scale of public dataset is limited, and proposed algorithms may be easier to learn special tricks (or patterns) due to lose good generalization. Serval methods (e.g. N-Beats [1]) that performed almost best on public datasets (e.g. M4 competition [2]), but performed very bad on our large-scale dataset. For reproducibility, we plan to release source code later after obtaining an open source license from our cooperation.
> [1]. Oreshkin, Boris N., et al. "N-BEATS: Neural basis expansion analysis for interpretable time series forecasting." arXiv preprint arXiv:1905.10437 (2019).
> [2]. https://www.kaggle.com/yogesh94/m4-forecasting-competition-dataset

---

> ### Author Response · Authors · 2020-11-20
> **Thank you very much for many insightful comments, and we plan to add more spatial baselines and discussion about linear attention.**
>
>
> 7. "Finally, I had issue with the Taylor expansion as its description was not clear (see detailed comments) but also I found description of related work missing in this aspect. Attention cost has been decreased to $N\sqrt{N}$, $N\log{N}$ and $N$ respectively in [3, 4, 5], this discussion is clearly missing to relate your contribution."
> Response: Thanks for your insightful comment. It should be noted that we cannot find the references "[3, 4, 5]" you quoted above. We try to compare several latest studies about linear attention acceleration and add this discussion in Section 3 later.
> The current attention acceleration methods can be roughly clarified into two groups. The first one is to construct kernel functions to approximate softmax. Katharopoulos et al. [1] constructs a kernel function with basis function $\varphi(x)=\phi(x)=elu(x)+1$. Shen et al. [2] replaces the calculation order of softmax, which is equivalent to the basis function $\phi(x)=softmax(x)$ and $\varphi(x)=e^x$.
> The second one is to modify attention's definition. For example, the time complexity of OpenAI's Sparse Attention [3] is close to linear, but its attention  hyperparameters are very hard to be initialized and actual efficiency is hard to ensure. The time complexity of Reformer [4] is $O(nlog(n))$. It uses LSH (Locality Sensitive Hashing) to find the maximum attention values quickly and then constructs a reversible FFN (Feedforward Network) to replace the original FFN and redesign back propagation process to reduce GPU memory usage. However, Reformer is difficult to implement and debug. Facebook’s Linformer [5] uses two additional matrices $E$ and $V$ to project $K$ and $V$, respectively, in order to get $Attention(Q,K,V)=softmax(Q(EK)^T )FV$. But the MLM experiment in Linformer needn't extract long-term dependence and cannot verify its linear time complexity. Moreover, many latest pooling methods (e.g., PoWER-BERT [6] and Funnel-Transformer [7]) are proposed to shorten feature maps but still difficult to implement and understand.
> [1]. Katharopoulos, Angelos, et al. "Transformers are rnns: Fast autoregressive transformers with linear attention." arXiv preprint arXiv:2006.16236 (2020).
> [2]. Shen, Zhuoran, et al. "Efficient Attention: Attention with Linear Complexities." arXiv preprint arXiv:1812.01243 (2018).
> [3]. Child, Rewon, et al. "Generating long sequences with sparse transformers." arXiv preprint arXiv:1904.10509 (2019).
> [4]. Kitaev, Nikita, Łukasz Kaiser, and Anselm Levskaya. "Reformer: The efficient transformer." arXiv preprint arXiv:2001.04451 (2020).
> [5]. Wang, Sinong, et al. "Linformer: Self-Attention with Linear Complexity." arXiv preprint arXiv:2006.04768 (2020).
> [6]. Goyal, Saurabh, et al. "PoWER-BERT: Accelerating BERT inference for Classification Tasks." arXiv preprint arXiv:2001.08950 (2020).
> [7]. Dai, Zihang, et al. "Funnel-Transformer: Filtering out Sequential Redundancy for Efficient Language Processing." arXiv preprint arXiv:2006.03236 (2020).
>
> 8. "3.1 at no point in the paper you say explicitly what is the loss that you are minimizing, this makes it harder to read. I would recommend to specify in 3.1 directly that you minimize NLE with poisson distribution (which is what I understood) and specify also clearly how predictions are made."
> Response: We employ quantile loss (with poisson distribution or not) to learn probabilistic distribution that introduced in Subsection 3.2 and Appendix A. We will move this part from appendix A to Subsection 3.1 to address your comment.
>
> 9. "3.3 the beginning of the section mix model description and related work"
> Response: The related work in Section 2 is all about probabilistic forecasting, but the beginning of the Subsection 3.3 is to explain the reasons that we employ "GAT" and "Transformer" as basic algorithms. We consider that they do not mix model description, otherwise readers may confuse that "why you choose GAT and Transformer".
>
> 10. 3.3 Eq. (3) has two unintroduced notation, what is the \bar? what is "batch"?
> Response: $\overline{[\cdot]}_{Batch}$ is the mean operator on the batch mode. We will add this notation in Subsection 3.3, and carefully verify other notations.

---

> > ### Comment · AnonReviewer3 · 2020-11-23
> > **missing references**
> >
> > I am sorry that references were not pasted correctly in my review, I have updated them.

---

> > > ### Author Response · Authors · 2020-11-23
> > > **Thanks for your references**
> > >
> > > Thanks for your references. We plan to do some experiments about existing linear attention methods to address this issue.

---

> ### Author Response · Authors · 2020-11-20
> **Thank you very much for many insightful comments, and we plan to add more spatial baselines and discussion about linear attention.**
>
> 1. "very hard to read, many details missing and notation are not introduced"
> Response: We are sorry for the lack of necessary details and notation descriptions, and we will revise a new version with more details.
>
> 2. "lack of relevant baselines (e.g. baselines using spatial information)"
> Response: We agree with your comment. Actually, our work does not focus on making full use of spatial information to greatly reduce forecasting error, but leveraging spatial information to improve training efficiency. However, to address your comment, we plan to do a series of experiments on spatial baselines  (e.g. ST-MGCN [1] and DMVST [2]).
> [1]. Geng, Xu, et al. "Spatiotemporal multi-graph convolution network for ride-hailing demand forecasting." Proceedings of the AAAI Conference on Artificial Intelligence. Vol. 33. 2019.
> [2]. Yao, Huaxiu, et al. "Deep multi-view spatial-temporal network for taxi demand prediction." In Thirty-Second AAAI Conference on Artificial Intelligence. 2018.
>
> 3. "lack of relevant metrics to illustrate the benefit of the contribution"
> Response: Maybe we have some different perspectives about your comment. In Appendix A, we have described Risk-q $\mathcal{R}_q$ as the metric of probabilistic forecasting, which is consistent with a series of compared SOTA studies (DeepAR, MQRNN, TFT, etc.).
>
> 4. "missing related work discussion for efficient attention computation"
> Response: Thanks a lot! We will include the related work about attention acceleration in Section 3. A simple comparison will be discussed in the 7th response.
>
> 5. "1) While the method claims ~5% error improvement on the ride-hailing benchmark, only one baseline have access to spatial information while numerous methods have been proposed to handle spatio-temporal forecast, only one baseline (TFT) is provided with spatial information (with no detail). "
>  Response: As mentioned in the second comment above, we plan to add more spatial baselines as competitors.
>
> 6. "2) Since you are interested in the probabilistic forecast of the joint demand/supply targets (collaborative), your experimental setup should consider a joint metric (CRPS-sum for instance [1,2]) rather than the average of R50/R90 metrics of demand/supply targets which is "blind" to correlation be"
> Response: Our collaborative supply and demand forecasting refers to the causal relationship (i.e., equation (1) and (2) ) but not continuous ranked probability score sum (CRPS-sum) between demand and supply. CRPS-sum is a popular metric for probability forecasting, but it is not what we actually need. Under the collaborative/causal supply and demand framework, we only need quantile probabilistic distribution to forecast responses, which is consistent with a series of compared SOTA studies (DeepAR, MQRNN, TFT, etc.) and aligned with our business situations.

---

> > ### Comment · AnonReviewer3 · 2020-11-23
> > **answer**
> >
> > Thank you very much for your detailed answer. However, I still have concerns regarding accepting the paper:
> >
> > - it is better now that are you are discussing prior work on efficient attention but there is no comparison in your experiments, past-work are only mentioned in the appendix (and unfortunately they were not discussed at all for the first version which is problematic given that efficient attention is a main contribution claim of the paper)
> >
> > - spatial-temporal baselines: I am glad that you added two baselines that have access to spatial information however the current comparison is still lacking in this aspect. A large number of methods have been proposed for spatio-temporal forecasting and your paper only compares with two. The benefit of the method already shrank from up-to 15% to up-to 9% (this should be updated in the abstract). The comparison here should be more extensive and proper details should be given on those baselines, it is unclear how much the claim improvement will shrink in light of those comparisons.
> >
> > - regarding my point about "lack of relevant metrics", I meant to highlight that you are only measuring error as if supply and demand were independent but they are clearly correlated (and you model predict correlated samples since the prediction of $y_v$ takes $x_v$ as input). Hence, it makes sense to have an error metric that measures this joint effect, CRPS-sum would be one possible example. This point is less problematic that the previous above though.
> >
> > Finally, regarding the technical clarity, I am glad that many typos were fixed but the changes are substantial (there were many technical issues in the submitted version) and I believe should be evaluated as another round of review.

---

> > > ### Author Response · Authors · 2020-11-23
> > > **Thanks for your useful comments again.**
> > >
> > > Thanks for your useful comments again. Before deadline of final submission, we would continue to improve more details and make this work easy to read.
> > >
> > > * We are grateful to find that efficient attention is an interesting and useful research field. We would try our best to compare prior work about this issue.
> > >
> > > * We completely agree that a large number of methods have been proposed for spatio-temporal forecasting, and we promise that we will add more experiments of SOTA spatial baselines. **Additionally, we would appreciate it if you could recommend several typical SOTA spatial baselines.** The maximum benefit of our method is still up-to 15% in Table 3, as we split Table 2 in first version into Table 2 and Table 3. The main hyperparameters of baselines are consistent with their original papers.
> > >
> > > * We agree with your comment. Based on equation (1) and (2), we first predict demand $x_v(t+1:t+\tau_{max})$, then predict supply $y_v(t+1:t+\tau_{max})$ given $x_v(t+1:t+\tau_{max})$ and other covariates. Forecasting order of both demand and supply reflects the causal relationship between them. Compared with joint effect (e,g. CRPS-sum), quantile risk losses (i.e., $R_{50}$ and $R_{90}$) are enough to evaluate the final performance and is fair to compare with SOTA peer-work.

---

### Official Review · AnonReviewer2 · 2020-10-28
**Spatio-temporal fusion transformers for predicting supply and demand in two-sided markets.**

**Rating:** 5
**Confidence:** 4

**Review:**

The authors propose an interpretable spatio-temporal fusion transformer for predicting supply and demand in ride-haling platforms.  More generally, the authors claim that their approach extends to other two-sided markets such as electric grids, retail etc by showing empirical results of their approach using data from these markets.  The paper is well-written and easy to follow.

The assumption that the supply is always conditioned on the demand x_v(, t + \tau_max) is too strong.  Is there a smoother of this assumption where the demand is dependent on a moving window over the past and future supply?  The experimental results show the efficacy of the proposed ML architecture.  However, one result that will greatly help the conclusions are results that clearly show the interpretability of the predictions, given that the authors state this as one of the main differences of their proposed solution.

The use of higher-order Taylor terms to approximate the attention procedure is interesting but the time complexity reductions are obvious and therefore does not meet the novelty bar for an ICLR submission.

Other suggestions - make the use of the word 'collaborative' in the model more clear.  The significance of the results is not completely clear and how the interpretability is helping understanding of the results.  Adding clarifications will help in the exposition.

I still have some concerns regarding the paper. The lack of baseline comparisons with spatio-temporal data (as also observed by fellow reviewers). My other concern also remains - from the authors' response, it is not clear how once can clearly attribute explainability of the results from their analysis of the model.

---

> ### Author Response · Authors · 2020-11-20
> **Thank you very much for insightful comments! We consider to add more descriptions about causal model ideas.**
>
> 1. "The assumption that the supply is always conditioned on the demand $x_v(, t + \tau_{max})$ is too strong. Is there a smoother of this assumption where the demand is dependent on a moving window over the past and future supply? "
> Response: Thanks a lot for your insightful comments! In practice, we choose different sliding windows to limit training time range. Specially, the sliding window is the past two weeks when forecast period is one day, and the sliding window is the past four weeks when forecast period is seven days. We will emphasize these settings in subsection 3.1, and the detailed experimental details will be listed in Appendix.
> In addition, we agree that the demand $x$ may depend on historical supply $y$ that happens several weeks (or even longer) ago. But in the last several weeks (i.e. training period), for customers, their demands may be primarily influenced by their own historical patterns and their recent request completion rates. Regarding to your question, we usually include historical request completion rate as an external covariate in $z$ to address demand forecasting perfectly. Moreover, we have included the historical and current demands to predict the future demands such that the proposed equations work reasonably well. Furthermore, we plan to introduce the request completion rate and integrate it with existing external covariates $z$.
>
> 2. "Other suggestions - make the use of the word 'collaborative' in the model more clear. The significance of the results is not completely clear and how the interpretability is helping understanding of the results. Adding clarifications will help in the exposition."
> Response: Thanks for your kindly suggestions. We will add further explanations about 'collaborative' in subsection 3.1 and more worthy details, such as discussion about causal attention and descriptions about experimental hyperparameters.

---

### Official Review · AnonReviewer4 · 2020-10-29
**Causal attention improves performance a lot.**

**Rating:** 6
**Confidence:** 3

**Review:**

This paper proposes a Transformer-like framework, named CausalTrans, to tackle the demand and supply problem in the ride-hailing market. The problem is formulated by training two probabilistic models which forecasts collaborative demand and supply, by given historical observations and dynamic covariates. The paper leveragesTransformer encoder-decoder architecture, and proposes submodule (Fast S.F., C.A. and T.A.) for different functionalities. Many experiments and ablation analysis are constructed to show that the proposed method outperforms state-of-the-art. The paper also provides good visualization regarding the casual attention model, facilitating to understand the proposed idea.

+ overall the paper is well organized and easy to read.
+ the proposed architecture is with merit: Fast S.F. with an approximate Taylor’s expansioninstead of using softmax function, showing lower computational cost with little performance dropped; C.A. is proposed to deal with HTE problem in large-scale spatio-temporal forecasting problems, where a DML algorithm is also proposed to learn the C.A. model.
+ the experimental results are comprehensive and promising.

Still, I have some questions:
- The ablation analysis shows that spatial fusion shows tiny improvement (+0.3% on average). I’d like the author to elaborate more on the reason. If there are hyperparameters for the spatial fusion, please do some ablation analysis in this regard.

- From experiments the fast attention improves computation a lot with only 0.2% performance dropped in multi-horizon methods. Is it the same in multi-horizon methods?

- From table 1 the results show that, in case of Electricity, the proposed method can’t outperform the state-of-the-art (TFT) due to lack of covariates and spatial information. I’d like to see the ablation analysis (like table 3) in case of Traffic, showing that in case of the iterative method, the performance can be gained by each submodule.

- Lack of experimental details. For example, learning rate/strategy, batch size, optimizer, architecture settings...etc. If the author(s) plan not to release the code in the future, it’s better to list the experimental details in the appendix, for reproducing the performance.

Overall I think that the proposed causal attention is valuable. By adapting the C.A. to the transformer architecture, the proposed method is comparable to state-of-the-art. Moreover, sufficient experiments demonstrate the proposed method can achieve decnet performance in the demand-supply problem. If the paper can better clarify the points mentioned above, I’ll vote for positive.

---

> ### Author Response · Authors · 2020-11-20
> **Thank you very much for insightful comments! We will add some ablation analysis in Appendix to address your comments.**
>
> 1. "The ablation analysis shows that spatial fusion shows tiny improvement (+0.3% on average). I’d like the author to elaborate more on the reason. If there are hyperparameters for the spatial fusion, please do some ablation analysis in this regard."
> Response: Thanks a lot! We feel that spatial fusion aggregates adjacent hexagonal grids, leading to reducing statistical noises in both demand and supply. For instance, in some cases, the boundary (usually around 800 meters) of adjacent grids separates large demand hotpots (e.g., large shopping malls), resulting in some noise when counting supply and demand. Spatial fusion can reduce the influence of such noise, while improving the probabilistic forecasting performance. According to the ablation analysis in Appendix C.2, the longer forecasting time (e.g., 7 days versus 1 day), the more significant gain by using spatial fusion. We consider the use of spatial fusion as a trick for enhancing the robustness of forecasting. The hyperparameter of spatial fusion is $\mathcal{K}$ used in the kmeans method. In this paper, we set $\mathcal{K} \in ${3, 4, 5}. To address your comments, we will add some ablation analysis in Appendix.
>
> 2. "From experiments the fast attention improves computation a lot with only 0.2% performance dropped in multi-horizon methods. Is it the same in multi-horizon methods?"
> Response: Thanks! It is an interesting discussion. We feel that, if attention feature maps (i.e. $Q$, $K$ and $V$ in original attention) can meet the positive definite and normalization conditions, and one focuses on short-term dependence, then the linear attention used here is useful for this aspect. The linear attention replaces softmax with the multiplication of feature maps, but such multiplication during the training is unstable with enforcing the positive definite and normalization conditions for feature maps. For our ride-hailing dataset, the positive definite condition is valid ($Q$, $K$ and $V$ are positive) and the L2 normalization is employed, so our training process is stable. In addition, almost all attentions with the linear complexity level (e.g., Sparse Attention proposed by OpenAI, Linformer proposed by Facebook, etc.) follow the same idea of "mainly retaining the attentions in short context and forcing most of attentions to zero". Moreover, linear attention in our work also focus on the short-term dependence such that the long-term dependence is ignored.
>
> 3. "From table 1 the results show that, in case of Electricity, the proposed method can’t outperform the state-of-the-art (TFT) due to lack of covariates and spatial information. I’d like to see the ablation analysis (like table 3) in case of Traffic, showing that in case of the iterative method, the performance can be gained by each submodule."
> Response: We will add a series of ablation experiments in case of Traffic.
>
> 4. "Lack of experimental details. For example, learning rate/strategy, batch size, optimizer, architecture settings...etc. If the author(s) plan not to release the code in the future, it’s better to list the experimental details in the appendix, for reproducing the performance."
> Response: We will include the detailed hyperparameters of the experiments (learning rate with decay strategies, optimizers, structure settings, etc.), training environment settings, training time and other details in Appendix. Moreover, we plan to release our source code later due to obtaining an open source license from our cooperation.

---

### Official Review · AnonReviewer5 · 2020-11-06
**Interesting paper, but some designs need more explanation.**

**Rating:** 6
**Confidence:** 2

**Review:**


This paper proposes a new framework of casual spatial-temporal prediction with high interpretability. Specifically, it first proposes a casual transformer including fast spatial graph fusion, casual attention, and temporal attention units. The authors conduct extensive experiments from different domains (electricity, traffic, etc.) and show the effectiveness, efficiency, and interpretability of their proposed method.

Strength:

+ The authors propose to reduce the complexity of the computation of the attention module.

+ The authors first propose a causal attention method for HTE in large-scale spatio-temporal prediction problems.

+ The experiments on datasets from various domains are adequate, which can support the authors' claim.

Weakness:

- Both equations (1) and (2) are based on the authors' assumptions. Do such assumptions have any support (either from previous literature or from the data)? In my opinion, some of the assumptions do not make sense, for example, one assumption is that $x_v(t + 1)$ is primarily affected by historical demands in $x_v(: t)$ and external covariates in $z$ without historical supply $y$. However, if the historical supply y is not enough, then the demand may raise because more and more demand accumulates.

- It seems that the authors' proposed approximate Taylor's expansion attention can be used for attention models or even softmax functions in any scenarios. Does it have any limitations? For example, some of the coefficients are very large so that the higher-order Taylor terms cannot be overlooked. Please explain why such problems do not exist in the spatial-temporal prediction scenario in this paper.

---

> ### Author Response · Authors · 2020-11-20
> **Thank you very much for all insightful comments! We plan to add more explanation and model details.**
>
> 1. "Both equations (1) and (2) are based on the authors' assumptions. Do such assumptions have any support (either from previous literature or from the data)? In my opinion, some of the assumptions do not make sense, for example, one assumption is that $x_v(t + 1)$ is primarily affected by historical demands in $x_v(: t)$ and external covariates in $z$ without historical supply $y$. However, if the historical supply y is not enough, then the demand may raise because more and more demand accumulates."
> Response: Thanks a lot for your insightful comments! These two equations are primarily based on our understanding of business. We completely agree with your two major comments. The first one is that the demand $x$ may depend on historical supply $y$ that happens several weeks (or even longer) ago. But in the last several weeks (training period), for customers, their demands may be primarily influenced by their own historical patterns and their recent request completion rates. The second one is that the demand may rise when supply is not enough for demand and demand may accumulate.
> In short, we feel that the distribution of demand is primarily determined by its historical distribution since the forecast period that we consider in the ride-sharing business is a short-term range (usually 1~7 days). Regarding to the first comment, we usually include historical request completion rate as an external covariate in $z$ to explain it better. The second one usually happens during the rush hour and lasts for 1-2 hours, leading to 2-4 outlying observations due to the temporal resolution being 30 minutes. Moreover, we have included the historical and current demands to predict the future demands such that the proposed equations work reasonably well. To address your comments, we plan to introduce the request completion rate and integrate it with existing external covariates.
>
> 2. "It seems that the authors' proposed approximate Taylor expansion attention can be used for attention models or even softmax functions in any scenarios. Does it have any limitations? For example, some of the coefficients are very large so that the higher-order Taylor terms cannot be overlooked. Please explain why such problems do not exist in the spatial-temporal prediction scenario in this paper."
> Response: Thank you so much for insightful comments! The linear attention designed in this paper essentially replaces softmax with the multiplication of feature maps (i.e. $Q$, $K$ and $V$ in original attention). However, such multiplication is unstable during the training. To address this issue, we need to ensure the positive definitive and normalization of feature maps. The feature maps in our ride-hailing dataset are positive definite ($Q$, $K$ and $V$ are positive), and L2 normalization is employed to ensure stable training processes. Additionally, almost all current linear complexity level attentions (e.g., Sparse Attention proposed by OpenAI, Linformer proposed by Facebook, etc.) follow the idea of "mainly retaining the attentions in short context and forcing most of attentions to zero". Linear attention in our work also mainly focus on the short-term dependence, since the long-term dependence that has little impact is ignored. To sum up, if the feature maps satisfy the positive definitive and normalization conditions, and we focuses on short-term dependence, then the linear attention method is a good option.

---

### Author Response · Authors · 2020-11-23
**Summary of revision**

We uploaded revisions of the submission in which

1. We add sentences to explain equation (1) and (2) in section 3.1. (done)
2. We move probabilistic forecasting in appendix to section 3.2. (done)
3. We add sentences to explain the motivation of using Taylor linear attention and correct equation (6) and (7) in section 3.4. (done)
4. We add experiments and analysis about two baselines ST-MGCN and DMVST in section 4.2, and we also check the comparison results in table 2 and table 3. (done)
5. We show experimental optimal hyperparameters in appendix B. (done)
6. We add ablation analysis on Traffic dataset in appendix C.2. (done)
7. We add ablation analysis about hyperparameter $k$ of spatial fusion in appendix C.2. (done)
8. We discuss a series of linear attention studies in appendix D. (done)

---

### Decision · Program_Chairs · 2021-01-07
**Final Decision**

**Decision:**

Reject

**Comment:**

The paper presents a spatial-temporal prediction framework with causal effects of predictors for better interpretability. The idea is interesting and the touch on modeling causal relations could be useful in practical applications. The paper receives mixed ratings and therefore there has been extensive discussion. We agree that while the paper has some merits, it falls short on the following aspects:

1, One central issue pointed out by all reviewers is the evaluation. For example, the contribution on efficient attention was not compared to any previous work; most of the baselines do no have access to the spatial information, which makes the comparison unfair. The authors did add two more baselines with access to spatial information. However, there are not enough details and discussions to make the results convincing; In addition, other stronger baselines should be added.

2. The notation and technical presentation was extremely lacking in the submitted version, the amount of unintroduced notations. Even in their core contribution equations had major issues with norm and vectors mixed together (see the difference between the corrected equation in the Taylor equation and the one in the original submission)

After the discussion, all reviewers agree that the paper fails to provide a fair and convincing evaluation, and the ratings will be adjusted to reflect the discussion. We hope that the reviews can help the authors improve the draft for a stronger submission in the future.